# Diagnostic Performance of Xpert MTB/RIF Ultra Compared with Predecessor Test, Xpert MTB/RIF, in a Low TB Incidence Setting: a Retrospective Service Evaluation

Mary Mansfield,[a] Anne Marie McLaughlin,[b] Emma Roycroft,[a,c] Lorraine Montgomery,[c] Joseph Keane,[b] Margaret M. Fitzgibbon,[a,c] Thomas R. Rogers[a,c]

[a]Department of Clinical Microbiology, Trinity College Dublin, St James's Hospital Campus, Dublin, Ireland
[b]Department of Respiratory Medicine, St. James's Hospital, Dublin, Ireland
[c]Irish Mycobacteria Reference Laboratory, St. James's Hospital, Dublin, Ireland

**ABSTRACT**   The aim of this study was to evaluate the performance of Xpert MTB/RIF Ultra (Ultra) compared with its predecessor, Xpert MTB/RIF (Xpert), in the diagnosis of tuberculosis (TB) in a low TB incidence country. Retrospective analysis was performed on 689 clinical samples received between 2015 and 2018, on which Xpert was performed, and on 715 samples, received between 2018 and 2020, on which Ultra was performed. Samples were pulmonary ($n = 830$) and extrapulmonary ($n = 574$) in nature, and a total of 264 were culture positive for *Mycobacterium tuberculosis* complex (MTBC). The diagnostic performance of both assays was analyzed using culture as the reference standard. The sensitivity of Ultra for culture positive (smear positive and smear negative) MTBC samples, was 93.2% (110/118) compared with 82.2% (120/146) for Xpert ($P = 0.0078$). In smear negative-culture positive samples, Ultra had a sensitivity of 74.2% (23/31) versus 36.11% (13/36) for Xpert ($P = 0.0018$). Specificity of both assays was comparable at 94.8% (566/597) for Ultra and 95.8% (520/543) for Xpert ($P = 0.4475$). The sensitivity of Ultra and Xpert assays among exclusively pulmonary samples was 95.3% (82/86) and 90.3% (84/93), respectively ($P = 0.1955$), and 87.5% (28/32) and 67.9% (36/53), respectively, among extrapulmonary samples ($P = 0.0426$). Ultra showed improved performance compared with Xpert in a low TB incidence setting, particularly in smear negative and extrapulmonary MTBC disease. The specificity of Ultra was lower than Xpert, however, this was not statistically significant.

**IMPORTANCE** The study demonstrates the improved sensitivity of the Ultra compared with the Xpert, particularly in smear negative TB disease, for both pulmonary and extrapulmonary samples in a low TB incidence setting. Cycle threshold (Ct) value for both assays was found to positively correlate with time to TB culture positivity, suggesting that Ct and semiquantitative results could be used as indicators of sample MTBC bacillary burden, and thus, perhaps, of transmission potential. This may have implications for the designation of patient isolation precautions.

**KEYWORDS**  Ireland, *Mycobacterium tuberculosis*, PCR, TB, Xpert MTB/RIF, Xpert MTB/RIF Ultra, diagnostic performance

**M**ycobacterium tuberculosis (Mtb) was responsible for approximately 1.4 million deaths and 10 million new infections globally in 2019 (1). The WHO estimates that only 71% of these new infections were identified and reported. This diagnostic gap is likely attributable, in part, to a lack of widespread access to sensitive and rapid diagnostic tests (2).

The Xpert MTB/RIF (Xpert) assay, produced by Cepheid (Cepheid, Sunnyvale, CA, USA), was endorsed by the WHO in 2010 for the rapid diagnosis of pulmonary tuberculosis and detection of rifampicin resistance (3). Xpert is an automated, integrated, real-

Address correspondence to Mary Mansfield, mansfime@tcd.ie.

The authors declare a conflict of interest. Parts of this service evaluation were submitted to Trinity College Dublin in the format of a Research Dissertation by Mary Mansfield for her MSc qualification and presented at the Irish Society of Clinical Microbiology Autumn Meeting 2021. Mary Mansfield was employed by Trinity College Dublin at the time of writing. Tom Rogers is an Advisory Board Honoraria for Insmed. This study was funded by the Irish Mycobacteria Reference Laboratory, St James's Hospital, and the Department of Clinical Microbiology, Trinity College Dublin.

time PCR, point of care assay that detects both the presence of *Mycobacterium tuberculosis* complex (MTBC) DNA and rifampicin resistance (RIF-R) associated mutations in the *rpoB* gene within 2 h (4). Results are given in semiquantitative categories; "high," "medium," "low," "very low," and negative. The Xpert has a limit of detection (LOD) of approximately 113 CFU per milliliter (CFU/ml), this is less sensitive than culture, which has an LOD of between 1 and 10 CFU/mL (5, 6).

The next generation assay, the Xpert MTB/RIF Ultra (Ultra), was endorsed by the WHO in 2017 to circumvent the comparatively lower sensitivity of the Xpert (7). Ultra has an LOD of 15.6 CFU/mL and uses PCR probes targeting multicopy sequences of IS6110 (16 copies/cell) and IS1081 (5 copies/cell) to detect the presence of MTBC DNA (6). The Ultra detects RIF-R by relying on the melting curves of sloppy molecular probes to detect mutations within the *rpoB* gene (6). Semiquantitative categories are the same as those used in the Xpert, but also include a new category; "trace."

Ireland is a low tuberculosis incidence country, with 5.6 cases per 100,000 population in 2019 (8). The Irish Mycobacteria Reference Laboratory (IMRL), St James's Hospital (SJH), Dublin, receives diagnostic samples from local and regional hospitals in Ireland. Samples from patients with suspected TB are investigated by smear microscopy, followed by mycobacterial culture on both solid and liquid media. The Ultra was introduced to the IMRL on the 1 April 2018, replacing the Xpert. The Ultra is routinely performed on all newly diagnosed smear-positive samples and is performed on smear-negative samples when requested by the clinical team.

The aim of our study was to evaluate the performance of the Ultra compared with its predecessor test, the Xpert, in the diagnosis of tuberculosis in a low TB incidence setting.

## RESULTS

**Comparative performance of Xpert and Ultra.** Of the 689 samples on which the Xpert was performed, 146 were culture positive for MTBC, of which 120 were Xpert positive. The overall sensitivity of the Xpert was 82.19% (120/146), and 36.11% (13/36) for exclusively smear-negative samples (Table 1). Of the 543 samples that did not grow MTBC (i.e., culture negative or nontuberculous mycobacteria [NTM] isolated), 23 were Xpert positive. All NTM samples (*n* = 55) were Xpert negative. The overall specificity of the assay was 95.76% (520/543). PPV and NPV were 83.92% and 95.24%, respectively (Table 1 and Fig. S1).

Of the 715 samples on which the Ultra was performed, 118 were culture positive for MTBC, 110 of which were Ultra positive, including 6 "trace" results. The overall sensitivity of the assay was 93.22% (110/118), with sensitivity of 74.2% (23/31) for exclusively smear negative samples (Table 1). Smear microscopy grade was not available for seven smear-positive MTBC specimens within the Ultra cohort. Of the 597 samples that did not grow MTBC (i.e culture negative or NTM isolated), 31 were Ultra positive, including 8 "trace" results. All NTM samples (*n* = 44) were Ultra negative. Specificity of Ultra was 94.81% (566/597), with overall PPV and NPV of 78.01% and 98.61%, respectively (Table 1 and Fig. S1).

**Pulmonary and extrapulmonary samples.** There were 405 pulmonary samples in the Xpert cohort and 425 in the Ultra cohort. There were 284 extrapulmonary samples in the Xpert cohort and 290 in the Ultra cohort. Sensitivity, specificity, PPV and NPV values for both sample types using either assay are summarized in Table 1.

**Correlation between semiquantitative results and time to culture positivity (TTP).** Median TTP decreased with increasing semiquantitative values for both assays (Fig. 1). For the Xpert group, significant differences in TTP were found between semiquantitative categories, with the exception of between "high" and "medium," "very low" and "low," and "very low" and "negative" semiquantitative categories, for which differences did not reach statistical significance.

For the Ultra cohort, significant differences in TTP were found between "high" and "low," "very low," "trace," and "negative" categories and between "medium" and "very low" and "negative" categories.

**TABLE 1** Comparative Performance of the Xpert and Ultra Assays

| Sample Type | Sensitivity % (95%CI)[a] | | | Specificity % (95%CI) | | | PPV % (95%CI) | | NPV% (95%CI) | |
|---|---|---|---|---|---|---|---|---|---|---|
| | Xpert | Ultra | P value | Xpert | Ultra | P value | Xpert | Ultra | Xpert | Ultra |
| All samples ($n = 1404$) | 82.19 (75.01–88.02) | 93.22 (87.08–97.03) | 0.0078 | 95.76 (93.71–97.30) | 94.81 (92.71–96.44) | 0.4475 | 83.92 (77.64–88.69) | 78.01 (71.51–83.38) | 95.24 (93.38–96.59) | 98.61 (97.31–99.28) |
| Smear positive ($n = 197$) | 97.27 (92.24–99.43) | 100 (95.85–100) | 0.1206 | | | | | | | |
| Smear negative ($n = 67$) | 36.11 (20.82–53.78) | 74.19 (55.39–88.14) | 0.0018 | | | | | | | |
| All pulmonary samples ($n = 830$) | 90.32 (82.42–95.48) | 95.35 (88.52–98.72) | 0.1955 | 97.44 (95.01–98.89) | 96.46 (93.90–98.16) | 0.471 | 91.30 (84.08–95.43) | 87.23 (79.64–92.27) | 97.12 (94.78–98.43) | 98.79 (96.91–99.53) |
| Smear positive pulmonary ($n = 150$) | 98.73 (93.15–99.97) | 100 (94.94–100) | 0.3415 | | | | | | | |
| Smear negative pulmonary ($n = 29$) | 42.85 (17.66–71.14) | 73.33 (44.9–92.21) | 0.0959 | | | | | | | |
| All extrapulmonary samples ($n = 574$) | 67.92 (53.68–80.08) | 87.5 (71.01–96.49) | 0.0426 | 93.51 (89.52–96.32) | 92.64 (88.74–95.51) | 0.7055 | 70.59 (58.72–80.20) | 59.57 (48.39–69.84) | 92.70 (89.56–94.96) | 98.35 (95.98–99.34) |
| Smear positive extrapulmonary ($n = 47$) | 93.55 (78.58–99.21) | 100 (79.41–100) | 0.2991 | | | | | | | |
| Smear negative extrapulmonary ($n = 38$) | 31.8 (13.86–54.87) | 75 (47.62–92.73) | 0.0086 | | | | | | | |

[a]95% CI: 95% confidence interval; Xpert: Xpert MTB/RIF; Ultra: Xpert MTB/RIF Ultra; PPV: positive predictive value; NPV: negative predictive value; n: total number of samples in both Xpert and Ultra cohorts combined.

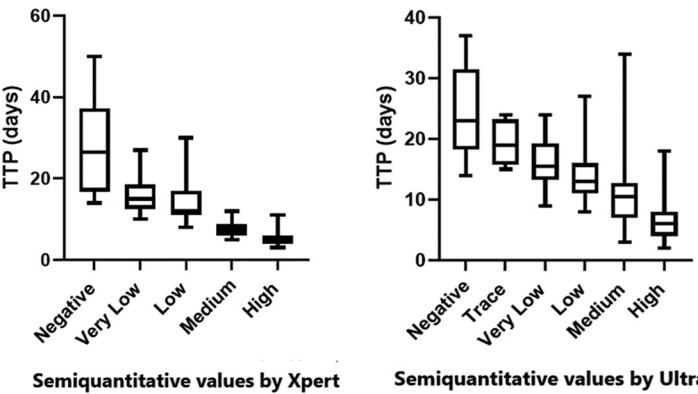

**FIG 1** Time to Culture Positivity versus Xpert and Ultra Assay Semiquantitative Result. Outliers removed using Rout method (Q = 1%). (Ultra 5/117, Xpert 8/146). Boxes represent 25th, 50th and 75th quartile of data with upper and lower whiskers representing minimum and maximum values. TTP: time to culture positivity.

**Correlation between semiquantitative result of Xpert and Ultra assays and smear microscopy.** All culture positive samples that were Ultra "trace" or negative were smear negative by microscopy. Among MTBC culture positive specimens, smear positive samples accounted for 100% of Ultra "high" results, 95.8% of Ultra "medium" results, and 65.4% and 56.3% of Ultra "low" and "very low" results, respectively. Eleven and a half percent of Xpert negative-culture positive samples were smear-positive (Table S1).

**Analysis of cycle threshold value.** Semiquantitative results from the Ultra and the Xpert are based on Ct values obtained from the binding of molecular probes to the *rpoB* gene. The earliest bound *rpoB* probe – that is to say, the lowest Ct value recorded of all bound probes, gives a quantitative estimation of bacillary burden and allows designation of semiquantitative result ("high," "medium," "low," "very low") (5). The lower the Ct for the first bound *rpoB* probe, the higher the TB bacillary burden.

Ct values for samples processed with the Xpert and Ultra assays were recorded, with the exception of assay-negative and Ultra "trace" positive samples, which do not yield a *rpoB* gene probe Ct value.

Xpert showed moderate positive correlation between Ct value for samples and TTP, with Spearman R of 0.7003 ($P < 0.0001$) (Fig. 2). Similar correlation was found between Ultra Ct value of a sample and TTP, with Spearman R of 0.6030 ($P < 0.0001$).

**Rifampicin resistance (RIF-R).** Four Xpert positive specimens showed RIF-R on Xpert, these results were concordant with phenotypic culture-based drug sensitivity testing (DST). One specimen with RIF-R on phenotypic DST was Xpert negative. This sample was smear negative with a TTP of 42 days, suggestive of a paucibacillary sample.

Within the Ultra cohort, the Ultra recorded 11 positive RIF-R results, 6 of which showed phenotypic culture-based RIF-R, the remainder of which had discordant culture results. Subsequent

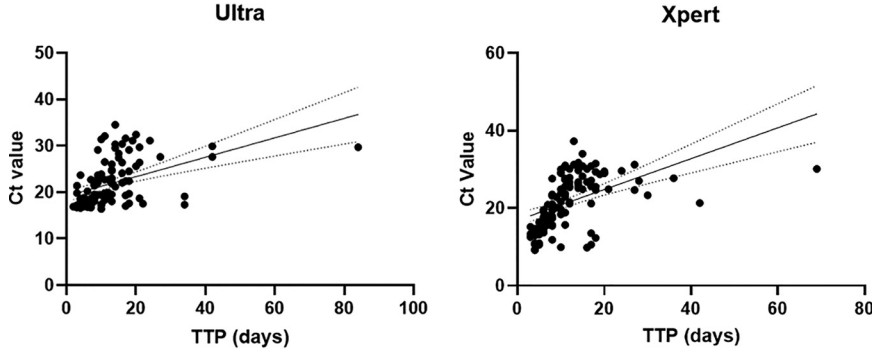

**FIG 2** Time to Culture Positivity versus Cycle Threshold for Xpert and Ultra Groups. Dotted lines represent 95% CI intervals. Ct: Cycle Threshold, TTP: time to culture positivity.

Sanger sequencing of *rpoB* gene and whole-genome sequencing revealed low confidence RIF-R associated mutations within these samples. Three were due to mutation A532V/A451V, as recently reported by Fitzgibbon et al. (9). The remaining two were due to Rif-R mutations, L511P/L430P and L533P/L452P, as previously described by Miotto et al. (10). These mutations have been described as "disputed," as the MICs of isolates harboring these mutations are close to the RIF critical concentration of 1.0 mg/L.

## DISCUSSION

Our study demonstrates the improved diagnostic sensitivity of the Ultra compared with the Xpert, particularly in smear negative TB disease, for both pulmonary and extrapulmonary samples. While we did observe a slight decrease in specificity of the Ultra compared with the Xpert, this did not reach statistical significance in any sample category.

The observed increase in sensitivity of the Ultra assay is consistent with findings from a number of diagnostic performance studies, in both high and low TB incidence settings (5, 11–13). A 2019 study from Switzerland on pulmonary samples, reported overall sensitivities of 83% and 95.7% for the Xpert and Ultra, respectively, and of 66.7% and 91.7%, respectively, for smear negative-culture positive samples. Xpert had a specificity of 97.3%, compared with 96.6% for the Ultra (11).

With regard to specificity, Dorman et al. first observed that the increased sensitivity associated with the Ultra assay came at the expense of a decrease in specificity, particularly among patients with a history of TB infection (13). This was replicated in several studies, notably in high TB burden settings (14, 15). Recently, a study from Cape Town, South Africa, found Ultra's specificity to be significantly less than that of Xpert (90% versus 99%), particularly among patients with a history of TB treatment within the preceding 2 years (69% versus 84%) (16).

Initial modeling studies by Kendall et al. suggested that, within high HIV and TB burden settings, the introduction of the Ultra was likely to result in a considerable mortality benefit. However, in areas of low prevalence, it may result in overdiagnosis and treatment (17). The modeling did not take into account the potential for variable specificity of the Ultra test depending on population incidence of TB and geographical location; however, the authors did note that this was a possibility. Subsequent studies in low incidence settings, have indicated a smaller decrease in specificity, compared with the Xpert, than previously observed in high burden settings (11, 18). While we observed that the specificity of the Ultra was consistently slightly lower than that of the Xpert, this did not reach statistical significance in any sample category.

The apparent difference in the Ultra's specificity between high and low TB incidence settings could, among other reasons, be due to overly sensitive detection of MTBC DNA from dead mycobacteria in patients previously exposed to TB within a high TB burden setting.

To mitigate the loss in specificity associated with the Ultra while maintaining sensitivity gains, studies have explored the effect of reclassifying Ultra "trace" results as negative (13, 14). In our setting, such reclassification would lead to a decrease in overall sensitivity from 93.2% (110/118) to 88.1% (104/118), with a substantial decrease in sensitivity among exclusively smear negative samples from 74.2% (23/31) to 54.8% (17/31). Associated specificity would increase slightly from 94.8% (566/597) to 96.1% (574/597). This highlights the added value of the "trace" call within our low TB incidence setting, particularly in paucibacillary samples.

TTP decreased with increasing semiquantitative result for each molecular assay. Other studies of the Ultra have also observed this effect (19–21). Ct values of both assays positively correlated with TTP. Our results suggest that Ct and semiquantitative results could be used as indicators of sample Mtb bacillary burden, and thus, perhaps, of transmission potential. This raises the question of whether the Ultra could replace smear microscopy for the designation of isolation precautions. Indeed, in South Africa, the Ultra has replaced smear microscopy as initial diagnostic test (22). Previous studies have supported the use of the Xpert in the designation of isolation precautions (23,

24), however, further studies would be useful to determine the role of the Ultra assay in this regard.

We observed an increase in sensitivity of the Ultra assay in detecting RIF-R, but also the presence of results which were discordant with phenotypic DST. A larger sample size would be needed for further analysis. The Ultra detects RIF-R by relying on the melting curves of 4 sloppy molecular probes to detect mutations within the *rpoB* gene. A "trace" positive result occurs when one or both of IS6110 and IS1081 probes are positive at Ct < 37 and one or less *rpoB* probes are positive at a Ct of <40, meaning that an interpretation of RIF-R cannot be made in such samples (7). This is a distinct drawback to the Ultra in settings with a preponderance of paucibacillary or Ultra "trace" positive samples.

A limitation to our study was that, due to its retrospective nature, two separate cohorts were analyzed rather than a side-by-side evaluation of both assays on each sample. Furthermore, 75.2% of samples were smear positive as, in our institution, molecular testing is routinely performed on newly diagnosed smear positive samples, while smear negative samples only undergo molecular testing if it is specifically requested by the clinical team. This high proportion of smear-positive pulmonary samples may not be reflective of the nature of TB disease within the community. Finally, no data on patient HIV status or history of previous TB infection was recorded.

## CONCLUSION

The Ultra assay represents a powerful diagnostic tool, showing improved sensitivity compared with the Xpert, particularly among smear negative TB specimens. While decreased assay specificity is a concern in areas of high TB prevalence, this decrease does not appear to be as pronounced within our low TB prevalence setting.

## MATERIALS AND METHODS

A retrospective analysis was performed on the records of 689 samples received at the IMRL between 2015 and 2018 on which the Xpert had been performed, and on 715 samples received between 2018 and July 2020 on which the Ultra had been performed. Molecular testing, with either the Xpert or Ultra assay, was performed on all newly identified smear-positive samples referred to the IMRL for TB culture. It was also performed on smear-negative samples when requested by the clinical team. Samples were pulmonary ($n = 830$) and extrapulmonary ($n = 574$) in nature, and a total of 264 samples were culture positive for *M. tuberculosis* complex. Of the culture positive MTBC samples, 197 (74.6%) were smear-positive and 67 (25.4%) were smear-negative.

For the purpose of this study, sputum, induced sputum and bronchoalveolar lavage samples were considered "pulmonary" samples. Extrapulmonary samples included a mixture of biopsy specimen samples, pleural fluid, CSF and lymph nodes. All sample types were decontaminated using 2% NaOH. Briefly, 0.5 mL of decontaminated sample was added to 1.5 mL Sample Reagent, mixed and incubated at room temperature for 15 min. Using a sterile Pasteur pipette, 2 mL of liquified sample was added to the Xpert or Ultra cartridge. All samples were cultured using the Bactec MGIT 960 culture system according to the manufacturer's instructions (Becton, Dickinson and Company, NJ, USA) and on Lowenstein Jensen slopes (E&O Laboratories, UK).

Data were extracted on smear microscopy result, Xpert or Ultra semiquantitative result, culture result and time to positivity (TTP) in liquid media for culture positive specimens. When positive, the cycle threshold (Ct) value from the earliest bound *rpoB* probe for each molecular assay was recorded. Samples were excluded from the study if assay results were "invalid" or "indeterminate," or, if culture result had not been recorded. TTP was not available for one culture positive sample in the Ultra cohort.

Statistical analysis was performed using GraphPad Prism version 9.0.2 and Medcalc. Sensitivity, specificity, positive and negative predictive values and 95% confidence intervals were evaluated using culture as reference standard. To compare the sensitivities and specificities of Xpert and Ultra assays, we used the Chi-square test. Differences were considered statistically significant if $P \leq 0.05$. The relationship between TTP and semiquantitative category was evaluated using Kruskal Wallis test and Dunn's test for multiple comparisons. The relationship between TTP and Ct value was assessed using Spearman correlation.

This service evaluation was approved for publication by the St James's Hospital Research and Innovation Office and by the St James's Hospital Data Protection Officer.

## SUPPLEMENTAL MATERIAL

Supplemental material is available online only.
**SUPPLEMENTAL FILE 1**, PDF file, 0.1 MB.

## ACKNOWLEDGMENTS

We acknowledge the work of all the laboratory staff in the IMRL involved in this study. We also acknowledge the many Irish hospitals, clinicians, and laboratory staff in addition to our own, for referring diagnostic specimens which generated the collection of samples used in this study.

Parts of this service evaluation were submitted to Trinity College Dublin in the format of a Research Dissertation by M.M. for her MSc qualification and presented at the Irish Society of Clinical Microbiology Autumn Meeting 2021. M.M. was employed by Trinity College Dublin at the time of writing.

This study was funded by the Irish Mycobacteria Reference Laboratory, St James's Hospital, and the Department of Clinical Microbiology, Trinity College Dublin.

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
