## [Reviewer comments · Microbiology Spectrum]

Microbiology Spectrum

Diagnostic performance of Xpert MTB/RIF Ultra® compared with predecessor test, Xpert MTB/RIF®, in a low TB incidence setting: a retrospective service evaluation.

Mary Mansfield, Anne Marie McLaughlin, Emma Roycroft, Lorraine Montgomery, Joseph Keane, Margaret Fitzgibbon, and Tom Rogers

Corresponding Author(s): Mary Mansfield, Trinity College Dublin

Review Timeline:

Submission Date:	November 29, 2021
Editorial Decision:	January 31, 2022
Revision Received:	April 3, 2022
Accepted:	April 6, 2022

Editor: Gyanu Lamichhane

Reviewer(s): The reviewers have opted to remain anonymous.

Transaction Report:

DOI: <https://doi.org/10.1128/spectrum.02345-21>

January 31, 2022

Dr. Mary Mansfield
Trinity College Dublin
Clinical Microbiology
Sir Patrick Dun Laboratory, St James's Hospital
James's Street
Dublin 8 D08RX0V
Ireland

Re: Spectrum02345-21 (Diagnostic performance of Xpert MTB/RIF Ultra® compared with predecessor test, Xpert MTB/RIF®, in a low TB incidence setting: a retrospective service evaluation.)

Dear Dr. Mary Mansfield:

Thank you for submitting your manuscript to Microbiology Spectrum. Two experts in the field of mycobacterial clinical microbiology have reviewed your manuscript and provided their feedback.

Link Not Available

Sincerely,

Gyanu Lamichhane, PhD
Editor, Microbiology Spectrum
Associate Professor
Division of Infectious Diseases
Johns Hopkins University School of Medicine

Journals Department
Reviewer comments:

Reviewer #1 (Comments for the Author):

The authors provide an evaluative comparison study of the two versions of GeneXpert (MTB/RIF and Ultra) to better understand diagnostic performance in a low TB incidence setting.

1. Findings from this study are comparative with previously published papers demonstrating the increased sensitivity for Ultra, with a slightly lower specificity. Some of these publications (with a 2019 publication by Pocognoli et al as example) have also shown these results in a low TB incidence setting. Given this overlap, it is suggested to the authors that additional details be added regarding potential impact of Ultra implementation in their region and the meaning and impact of "Trace" calls (which needs to be further expounded upon) to further the impact of test data amidst previous publications.

2. The difference in Xpert negative vs Ultra negative samples is striking (11.5% vs 0%), and should be discussed.
3. Time to positivity (TTP) can differ between liquid and solid media, this should be expounded upon.
4. Note that in some countries (South Africa, for example), Xpert Ultra has replaced smear microscopy as an initial test. This should be added to the discussion.
5. Line 148: Change to Cape "Town"
6. Line 227: Change "Fitzgibbon"

Reviewer #2 (Comments for the Author):

In this manuscript, "Diagnostic performance of Xpert MTB/RIF Ultra® compared with predecessor test, Xpert MTB/RIF®, in a low TB incidence setting: a retrospective service evaluation", the authors, Mary Mansfield et al., show that Xpert MTB/RIF Ultra has a higher sensitivity but lower specificity than the previous method Xpert MTB/RIF in detecting *M. tuberculosis* in pulmonary and extrapulmonary samples. The phenotypic DST did not detect resistance potentially caused by some *rpoB* mutations; this should have been discussed. The manuscript was rather well structured but would have gained on being shorter.

#1 Line 1 (Abstract) and Line 49 (Introduction): The words "MTBC infection" is typically used for latent TB, not for active tuberculosis, which is diagnosed with the methods described in the manuscript.

#2 Methods Lines 51 - 56: I miss a brief description of the methods used, both for Xpert and Ultra and for culture. How were sample types not mentioned in the Xpert/Ultra Cepheid's Instructions for Use treated? How were samples with small specimen volumes treated? As far as I can understand, the study was done on all routine samples 2015 -- 2020 positive or negative for MTBC in culture and had an Xpert or Ultra result; however, this is not stated. Which samples types were/were not analyzed routinely?

#3 Methods Lines 60 - 64: How many samples were "invalid"? Were any other samples excluded, except those mentioned in the text? How were microscopy-positive samples with the growth of non-tuberculous mycobacteria handled in the study?

#4 Results: Figure 1 content is not mentioned anywhere in the text. Thus, it is unclear what Figure 1 is supposed to illustrate what cannot be read in Table 1.

#5 Results Lines 93 - 99: I find it challenging to understand by the text which semiquantitative values are significantly different from TTP. Maybe this could be illustrated in Figure 2?

#6 Figure 2 would, in my view, look better if both y-axes had the same scale and each category on both x-axes were equally broad. Indicate, if possible, the significant differences.

#7 Results Lines 108 - 109: How many smear-positive samples were non-tuberculous mycobacteria?

#8 Figure 3 would, in my view, look better if both y-axes had the same scale and the x-axes had the same scale

#9 Results Line 113: The mutations are not borderline but critical concentration for the phenotypic resistance testing of rifampin.

#10 Discussion, Lines 145 - 150: Please speculate what you think the causes for the differences in specificities between Xpert and Ultra might be.

#11 Discussion, Lines 151 - 159: Please discuss why the differences in specificities between Xpert and Ultra are less in your low incidence setting than in a high incidence setting.

#12 Discussion, Lines 167 - 171: Discuss why Ultra cannot detect RIF-R in trace positive samples. Line 171: Should not all cultures undergo DST, regardless of Xpert/Ultra RIF result? The detection of RIF-R described in the Results section should be discussed.

#13 Discussion, Lines 172 - 178: You have observed a Ct overlap between different semiquantitative categories. What is the reason for that overlap? Is this overlap important at all? If it is essential, what is the use of semiquantitative categories?

Staff Comments:

Preparing Revision Guidelines

Please return the manuscript within 60 days; if you cannot complete the modification within this time period, please contact me. If you do not wish to modify the manuscript and prefer to submit it to another journal, please notify me of your decision immediately so that the manuscript may be formally withdrawn from consideration by Microbiology Spectrum.

Reviewer #1 (Comments for the Author):

The authors provide an evaluative comparison study of the two versions of GeneXpert (MTB/RIF and Ultra) to better understand diagnostic performance in a low TB incidence setting.

1. Findings from this study are comparative with previously published papers demonstrating the increased sensitivity for Ultra, with a slightly lower specificity. Some of these publications (with a 2019 publication by Pocognoli et al as example) have also shown these results in a low TB incidence setting. Given this overlap, it is suggested to the authors that additional details be added regarding potential impact of Ultra implementation in their region and the meaning and impact of "Trace" calls (which needs to be further expounded upon) to further the impact of test data amidst previous publications

Please see lines 176-182.

2. The difference in Xpert negative vs Ultra negative samples is striking (11.5% vs 0%), and should be discussed.

Wording clarified line 119-123. This reflects improved sensitivity of Ultra and its ability to detect MTBC even in smear negative samples.

3. Time to positivity (TTP) can differ between liquid and solid media, this should be expounded upon.

TTP was determined from growth on liquid media. Line 68-70.

4. Note that in some countries (South Africa, for example), Xpert Ultra has replaced smear microscopy as an initial test. This should be added to the discussion.

Please see line 188.

5. Line 148: Change to Cape "Town"

Amended.

6. Line 227: Change "Fitzgibbon"

Amended.

Reviewer #2 (Comments for the Author):

In this manuscript, "Diagnostic performance of Xpert MTB/RIF Ultra® compared with predecessor test, Xpert MTB/RIF®, in a low TB incidence setting: a retrospective service evaluation", the authors, Mary Mansfield et al., show that Xpert MTB/RIF Ultra has a higher sensitivity but lower specificity than the previous method Xpert MTB/RIF in detecting M. tuberculosis in pulmonary and extrapulmonary samples. The phenotypic DST did not detect resistance potentially caused by some rpoB mutations; this should have been discussed. The manuscript was rather well structured but would have gained on being shorter.

Amended.

#1 Line 1 (Abstract) and Line 49 (Introduction): The words "MTBC infection" is typically used

for latent TB, not for active tuberculosis, which is diagnosed with the methods described in the manuscript.

Amended.

#2 Methods Lines 51 - 56: I miss a brief description of the methods used, both for Xpert and Ultra and for culture. How were sample types not mentioned in the Xpert/Ultra Cepheid's Instructions for Use treated?

Details of methods added. All samples (both pulmonary and extra-pulmonary) were processed the same way for Xpert and Ultra tests.

How were samples with small specimen volumes treated? All sample types were decontaminated prior to processing on the Xpert assays. 0.5 ml of the decontaminated sample was added to 1.5 ml of Sample Reagent and added to the cartridge.

As far as I can understand, the study was done on all routine samples 2015 -- 2020 positive or negative for MTBC in culture and had an Xpert or Ultra result; however, this is not stated. Which sample types were/were not analyzed routinely?

Please see lines 57-79.

#3 Methods Lines 60 - 64: How many samples were "invalid"?

I am unsure of exact number.

Were any other samples excluded, except those mentioned in the text?

Lines 74-75

How were microscopy-positive samples with the growth of non-tuberculous mycobacteria handled in the study?

Lines 91-92 and Lines 99-100.

#4 Results: Figure 1 content is not mentioned anywhere in the text. Thus, it is unclear what Figure 1 is supposed to illustrate what cannot be read in Table 1.

I have moved this to the supplementary section. The figure represents a graphical representation using ROC curves of the PPV/NPV data which the journal's readers may find helpful when read in conjunction with the data shown in Table 1.

#5 Results Lines 93 - 99: I find it challenging to understand by the text which semiquantitative values are significantly different from TTP. Maybe this could be illustrated in Figure 2?

Adding extra notations to the graph made it difficult to read.

#6 Figure 2 would, in my view, look better if both y-axes had the same scale and each category on both x-axes were equally broad. Indicate, if possible, the significant differences.

Agreed, but when inputting the data into GraphPad Prism, due to the significant difference in ranges between the 2 data sets, the graphs were difficult to compare to each other.

#7 Results Lines 108 - 109: How many smear-positive samples were non-tuberculous mycobacteria?

Total NTM included in lines 86 and 94.

#8 Figure 3 would, in my view, look better if both y-axes had the same scale and the x-axes had the same scale

See answer to # 6.

#9 Results Line 113: The mutations are not borderline but critical concentration for the phenotypic resistance testing of rifampin.

Removed "borderline" and added sentence to explain. Lines 146-147.

#10 Discussion, Lines 145 - 150: Please speculate what you think the causes for the differences in specificities between Xpert and Ultra might be.

Please see line 173-175.

#11 Discussion, Lines 151 - 159: Please discuss why the differences in specificities between Xpert and Ultra are less in your low incidence setting than in a high incidence setting.

Please see lines 173-182.

#12 Discussion, Lines 167 - 171: Discuss why Ultra cannot detect RIF-R in trace positive samples.

Please see lines 193-196.

Line 171: Should not all cultures undergo DST, regardless of Xpert/Ult 1ra RIF result?

Agreed – comment deleted.

The detection of RIF-R described in the Results section should be discussed.

Please see lines 191-192.

#13 Discussion, Lines 172 - 178: You have observed a Ct overlap between different semiquantitative categories. What is the reason for that overlap? Is this overlap important at all? If it is essential, what is the use of semiquantitative categories?

Following discussion with Cepheid – overlap explained by change in assay version and so relevant sections removed.

April 6, 2022

Dr. Mary Mansfield
Trinity College Dublin
Clinical Microbiology
Sir Patrick Dun Laboratory, St James's Hospital
James's Street
Dublin 8 D08RX0V
Ireland

Re: Spectrum02345-21R1 (Diagnostic performance of Xpert MTB/RIF Ultra® compared with predecessor test, Xpert MTB/RIF®, in a low TB incidence setting: a retrospective service evaluation.)

Dear Dr. Mary Mansfield:

Your manuscript has been accepted, and I am forwarding it to the ASM Journals Department for publication. You will be notified when your proofs are ready to be viewed.

Sincerely,

Gyanu Lamichhane, PhD
Editor, Microbiology Spectrum
Associate Professor
Division of Infectious Diseases
Johns Hopkins University
